evolution, developmental biology

*Anolis* lizards, adaptive radiation, allometries, diversification, limb morphology, macroevolution

**Author for correspondence:**
Nathalie Feiner
e-mail: nathalie.feiner@biol.lu.se

# A highly conserved ontogenetic limb allometry and its evolutionary significance in the adaptive radiation of *Anolis* lizards

Nathalie Feiner, Illiam S. C. Jackson, Eliane Van der Cruyssen and Tobias Uller

Department of Biology, Lund University, Sweden

NF, 0000-0003-4648-6950; ISCJ, 0000-0002-7948-2860; EVdC, 0000-0003-4931-8880;
TU, 0000-0003-1293-5842

Diversifications often proceed along highly conserved, evolutionary trajectories. These patterns of covariation arise in ontogeny, which raises the possibility that adaptive morphologies are biased towards trait covariations that resemble growth trajectories. Here, we test this prediction in the diverse clade of *Anolis* lizards by investigating the covariation of embryonic growth of 13 fore- and hindlimb bones in 15 species, and compare these to the evolutionary covariation of these limb bones across 267 *Anolis* species. Our results demonstrate that species differences in relative limb length are established already at hatching, and are resulting from both differential growth and differential sizes of cartilaginous anlagen. Multivariate analysis revealed that Antillean *Anolis* share a common ontogenetic allometry that is characterized by positive allometric growth of the long bones relative to metapodial and phalangeal bones. This major axis of ontogenetic allometry in limb bones deviated from the major axis of evolutionary allometry of the Antillean *Anolis* and the two clades of mainland *Anolis* lizards. These results demonstrate that the remarkable diversification of locomotor specialists in *Anolis* lizards are accessible through changes that are largely independent from ontogenetic growth trajectories, and therefore likely to be the result of modifications that manifest at the earliest stages of limb development.

## 1. Introduction

Morphological diversification of ecologically specialized forms is a hallmark of adaptive radiations. The morphological differences that accrue are often substantial, but comparison of adult phenotypes reveals that even extreme cases of diversification tend to preserve covariation between characters [1–3]. One explanation for this is that the phenotypic variation that selection can act upon is highly structured by development [4]. Developmental integration can be observed and quantified in terms of the covariation between morphological characters, such as the bones of the limb, as they develop and grow [5–9]. If the regulation of development and growth were to exercise a substantial effect on adaptive diversification, distantly related species should share similar patterns of morphological covariation in ontogeny, and this may force adult morphologies to diverge along the same dimensions (i.e. developmental lines of least resistance [4,10]).

While an increasing number of studies compares trait covariation across species (e.g. [1,11–14]), and sometimes relate these to ontogenetic trajectories [6,15,16], little has been done in the context of adaptive radiations. Thus, the extent to which ontogenetic covariation of morphological characters evolves, as well as the extent to which it contributes to the pattern of morphological diversification across the phylogeny, remains poorly understood. One outstanding opportunity to investigate these issues is the adaptive radiation of *Anolis* lizards, a highly diverse clade of almost 400 species that demonstrate extensive morphological diversification. While roughly two-thirds of all *Anolis* lizards are

**Figure 1.** Phylogenetic relationship of species included in this study and selected adult phenotypes. Phylogenetic relationships (*a*) are adapted from Poe *et al.* [26]. The 15 species that are comprised in the ontogenetic dataset are highlighted with their scientific names. Circles on tree tips indicate ecomorph classifications, and biogeographic groups are marked by colour-coded arcs. Sample sizes of each species in the ontogenetic dataset are given in brackets after species names. Adult phenotypes are shown for six *Anolis* species (*b*). Note that all individuals are perching on the same stick to convey relative differences in body size. (Online version in colour.)

native to the mainland of Central and Southern America, many species are found on the islands of the Greater and Lesser Antilles. The match between morphology and micro-habitat, best documented on the four Greater Antillean islands (Cuba, Hispaniola, Puerto Rico and Jamaica), is often spectacular and strongly pronounced in limb morphology, in particular hindlimb length [17,18]. For example, species with long limbs tend to be agile runners on tree trunks, whereas species with short limbs tend to be specialized for perching on twigs and branches [19,20]. This match is captured by the ecomorph classification, which recognizes up to six habitat specialist types (ecomorphs) that have repeatedly and convergently evolved on the four Greater Antillean islands [21–23]. However, not all Greater Antillean species are classified as ecomorphs and roughly 10% exhibit more unique morphologies. While limb morphology is arguably the most defining ecological feature of *Anolis*, the developmental basis of adaptive variation has received little attention. One study comparing limb ontogenetic growth between pairs of *Anolis* species belonging to different ecomorphs found that relative limb length differences are established very early in development and that differential growth contributes little, if anything, to ecomorph-specific differences [24]. Another study on two *Anolis* species confirmed the early appearance of species-specific limb length differences, but found that these increase over developmental time, even after hatching [25]. Besides this question of which period of limb development that is evolutionarily labile, the extent to which growth trajectories have shaped evolutionary diversification of the *Anolis* adaptive radiation is yet to be addressed.

Here, we fill these gaps in knowledge by investigating the covariation of 13 limb bones during embryonic development of 15 species, and subsequently comparing the ontogenetic covariation in these species against patterns of diversification

of the same limb bones across adults of 267 island and mainland *Anolis* species. The 15 species of the ontogenetic series include eight species of four ecomorphs and seven non-ecomorph species, all native to the Greater or Lesser Antilles (figure 1). Note that the ontogenetic series included only developmental stages up until hatching, but no juvenile stages. Using these datasets, we addressed the following questions. Firstly, we tested if species-specific relative limb length is established already at hatching, or even earlier, and if these differences emerge through differential growth rate of individual bones or differential size of limb anlagen. Secondly, we used multivariate ontogenetic allometry to assess if the covariation of limb bones during embryonic development is conserved between species. Thirdly, we applied the same approach to an equivalent morphological dataset comprising adult limb data of 267 species and assessed if evolutionary covariation is conserved between the three major biogeographic groups. Finally, we tested if the major axis of embryonic bone growth is aligned with evolutionary allometry, as would be expected if growth trajectories represented a developmental line of least resistance during adaptive radiation.

## 2. Material and methods

### (a) Ontogenetic dataset

We established breeding groups for 15 *Anolis* species at Lund University. *A. sagrei* were collected at Palm Coast, Florida and *A. equestris* in Miami, Florida in April 2016 and brought to the animal facility at Lund University. The remaining species were purchased through the commercial pet trade or private breeders. Each breeding group consisted of one male and two females (except for *A. porcus* that consisted of a single female) and a minimum of one breeding group per species was established. Since

our aim was to compare embryonic development across species, individual variation within species was deemed sufficiently small to be neglected in this study.

Breeding groups were housed in 80 l plastic cages (Wham Crystal box with mesh on top, 590 × 390 × 415 mm), except for *A. equestris* and *A. porcus* that were kept in Zoo Med Reptibreeze open air screen cages (610 × 610 × 1200 mm). During breeding, lizards were kept at a light cycle of 12 L : 12 D and given access to basking lights (60 W) for 10 h per day and a UV light (Exo-Terra 10.0 UVB fluorescent tube) for 6 h per day. Live food (mealworms, crickets, wax worms, cockroaches (only for *A. equestris*) and snails (only for *A. porcus*)) were provided ad libitum. Eggs were collected every day and incubated at 26°C in individual small plastic containers filled two-thirds with moist vermiculite (5 : 1 vermiculite:water volume ratio) and sealed with clingfilm. While different *Anolis* species probably have different optimal incubation temperatures [27], 26°C is a good compromise for the species included in the present work.

To first establish the incubation time for each species, at least one embryo was allowed to hatch (electronic supplementary material, table S1). On the basis of this species-specific incubation time, embryos were dissected at 12 evenly spaced time points from oviposition to hatching to obtain an ontogenetic series of each species that provides a broad overview of the variation of morphogenesis (electronic supplementary material, figure S1). Eggs were dissected in phosphate-buffered saline (PBS) and embryos were sacrificed, fixed in 4% paraformaldehyde for 1–4 days and stored in 100% methanol. Embryos were staged according to the *Anolis* staging table [28]. To accurately measure individual limb bones, embryos were subjected to a clearing and staining procedure [29]. We found that this procedure produced satisfactory results for embryos from stage 11 onwards and therefore excluded younger stages. Stained limbs (dissociated from the rest of the body) were mounted horizontally on a petri-dish and photographed under a dissection microscope (Olympus SZX10). ImageJ [30] was used to record measurements of the limb bones (in millimetre to the closest 0.001 mm) that span the maximal proximo-distal axis of both fore- and hindlimbs. These include the long bones (humerus, ulna, femur and tibia), metapodials (metatarsus and metacarpal), and the phalanges of the longest digit of both fore- and hindlimb (electronic supplementary material, figure S2). In total, we recorded six individual bone elements for the forelimb, and seven elements for the hindlimb. The body posture of embryos makes the common measure of body size in lizards, snout-vent length, prone to measurement error. Therefore, we recorded a dorsal view of the flattened embryo to obtain a measure of the length from the first cervical vertebra to the sacral vertebra, an estimate of body size that we term spine length. These measurements were highly repeatable (Pearson's product-moment correlation $r_P = 0.980$, $n = 47$). A small number ($n = 253$; 2.95%) of individual measurements were missing due to fractured bones, and we imputed these missing values using the 'pcaMethods' package [31] based on all linear measurement per species.

## (b) Evolutionary dataset

Measurements of adult limb elements were obtained from micro-CT scanning of museum specimens that were published elsewhere [12,13]. In brief, specimens were scanned using a GE phoenix v | tome | x m system (source voltage 100 kV; source current 200 μA; isometric voxel size 55–75 μm) at the Nanoscale Facility of the University of Florida, US. Reconstructed image stacks (software GE phoenix datos | x CT) were further processed using VGStudio MAX software (v. 3.2) by applying manual thresholding to extract surface models of skeletal structures. Linear measurements were directly obtained using the VGStudio MAX software. For each lizard, we measured the same limb elements as detailed above

(to the closest 0.01 mm). This was achieved by placing one landmark each on the proximal and on the distal end of a bone and extracting the distance between these two points in 3D space. These measurements were highly repeatable (Pearson's product-moment correlation $r_P = 0.992$, $n = 40$). A small number ($n = 51$; 0.47%) of individual measurements were missing due to fractured bones, and we imputed these missing values using the 'pcaMethods' package [31] based on all linear measurements of all individuals. Estimates of body size were inferred from the centroid size of the hip girdles and were shown to be highly correlated with snout–vent length (see Methods in [12]).

## (c) Statistical analyses

All limb elements of the ontogenetic and the phylogenetic dataset were log-transformed prior to analyses. Linear models were used to assess the relationship between total limb length and spine length, and to test if species differ in their relative limb length at early embryonic stages, at hatching and as adults. The relative limb length for fore- and hindlimbs was captured by the sum of their respective limb bones divided by spine length for embryos and hatchlings, and divided by centroid size for adults. We used ANCOVA and MANCOVA to test for species differences in the bivariate allometry between limb length (total or individual bones) and spine length.

Multivariate allometry, the relative proportions of individual limb bones in relation to overall size, was quantified for fore- and hindlimbs separately, using a principal component analysis (PCA) approach [32]. Determining the dimension in multivariate space that maximizes variation, the patterns of covariation in a given dataset can be quantified and this approach is commonly used for studying ontogenetic, static and evolutionary allometries [33,34]. To test if species differed in their multivariate ontogenetic allometric slopes, we performed principal component analyses on log-transformed bone measurements for each species. The coefficients of the first principal components (i.e. the allometric vectors) serve as an indicator of allometry for each limb element and is effectively a measure of ontogenetic covariation between limb elements [33].

We used these coefficients in three ways. First, as observations in a second PCA that resulted in the construction of an 'allometric space' [15,35], which visually presents patterns of multivariate allometry. Second, coefficients were used to test if individual limb elements scale isometrically with size. If all individual limb elements would contribute equally to the increase in size over ontogeny, the coefficients of the allometric vector would be equal to $p^{-0.5}$ where $p$ is the number of elements [34]. We applied non-parametric bootstrapping (1000 iterations) of the dataset to test if the range of resampled coefficients includes the value expected under multivariate isometry (coefficient = $p^{-0.5}$). Third, coefficients were used as vectors in multivariate space that can be compared between pairs of species to extract an angle (arc cosine of the inner product of the two vectors) that describes how well two ontogenetic allometries are aligned [33]. To assess whether or not the observed angles between pairs of species were larger than expected if all species share a common allometric pattern, we created a dataset that conformed to a shared allometric pattern and applied non-parametric bootstrapping to generate a distribution of test statistics [32] (see electronic supplementary material for extended methods).

To test if the ontogenetic and phylogenetic allometries were aligned, we compared the common ontogenetic allometric vector with the evolutionary allometric vector of the same major group. The *Anolis* clade originated on the mainland, but has colonized Caribbean islands in two waves—one to the Southern Lesser Antilles, and one to the Greater and Northern Lesser Antilles—and the latter gave rise to a re-colonization of the mainland. Since the Southern Lesser Antillean clades contain only few

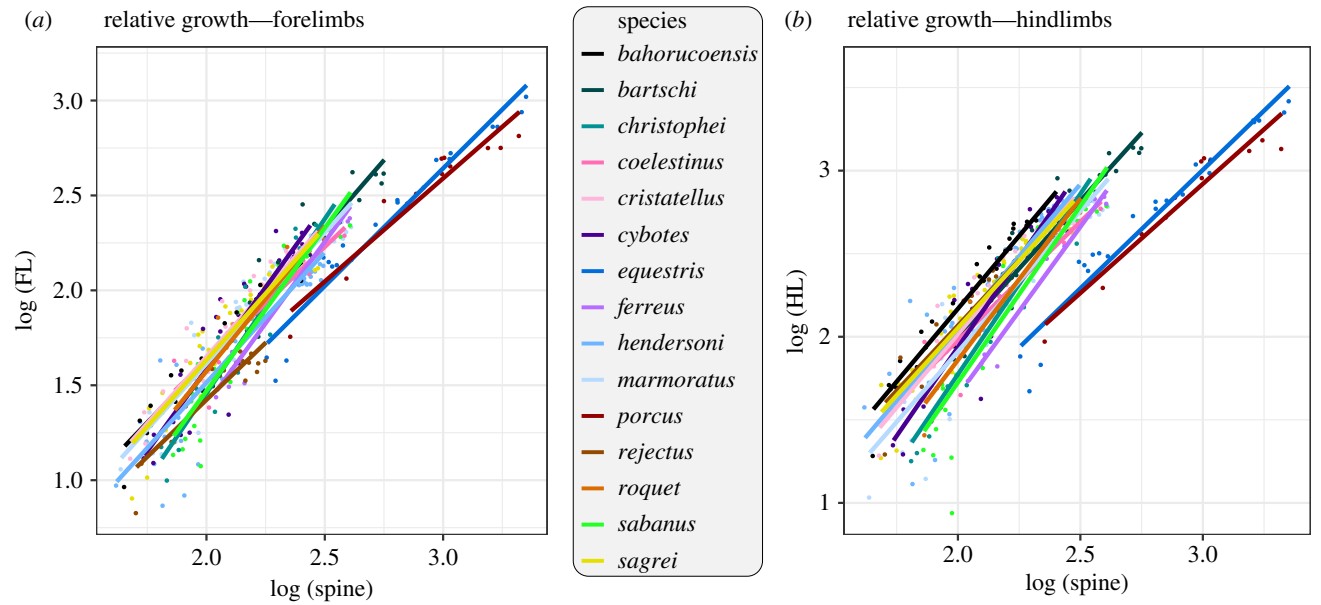

**Figure 2.** Patterns of bivariate allometry across 15 *Anolis* species. Regressions of (*a*) fore- and (*b*) hindlimbs against spine length are plotted for each of the 15 *Anolis* species. The two species with exceptionally large body size and relatively short limbs are *A. equestris* and *A. porcus*. (Online version in colour.)

species, we divide *Anolis* lizards into three major groups. The first two include the group that gave rise to an independent diversification on the mainland (the Primary Mainland group, MLpri; 29 species in our dataset) and the group that diversified on the mainland following a re-colonization from the Greater Antilles (the Secondary Mainland group, MLsec; 107 species in our dataset). The remaining group includes all the Greater Antillean (110 species in our dataset) and the Northern Lesser Antillean *Anolis* (11 species in our dataset). We refer to this group as the Antillean group (Ant). One species in our embryonic dataset, *A. roquet*, is native to the Southern Lesser Antilles and does not belong to any of the three major groups, and we therefore excluded this species from the ontogenetic dataset for this analysis. We assessed the alignment of the common ontogenetic allometric vector (derived for the remaining 14 species) to the evolutionary allometric vector of their native Antillean group (Ant), and to the two mainland groups (MLpri and MLsec).

Evolutionary allometric vectors (pPC1) were derived from phylogenetic PCAs in the R package 'phytools' [36] for each of the three major groups based on species means of each measurement and the phylogenetic tree published by Poe *et al.* [26] with modifications described in [13]. We first assessed how well the phylogenetic allometric vectors of the three major groups are aligned with each other. Secondly, we calculated the angle between the common ontogenetic allometric vector (cPC1) and the evolutionary allometric vectors (pPC1) of each major *Anolis* group (observed angles $\Theta_{ont-Ant}$, $\Theta_{ont-MLpri}$, $\Theta_{ont-MLsec}$) and assessed their statistical significance by bootstrapping from rotated datasets (see electronic supplementary material for extended methods). Since we compared ontogenetic and phylogenetic major axis that were derived by slightly different methodologies (common PCA versus phylogenetic PCA), we repeated all analyses that involve phylogenetic PCAs and replaced these with standard PCAs and accordingly performed non-parametric bootstrapping to generate the sampling distributions. This alternative approach produced qualitatively identical results that are presented in the electronic supplementary material. All statistical analyses were conducted in R (v. 3.6.1) [37].

## 3. Results

The final dataset comprised linear morphometric measurement of 374 individual embryos and hatchlings belonging

to 15 *Anolis* species (mean, 24.93 individuals per species ± 12.99 standard deviations, s.d.) and 693 individual adults belonging to 267 *Anolis* species (mean, 2.60 individuals per species ± 1.58 s.d.). Eleven species of the embryonic dataset belong to the Greater Antillean group, three to the Northern Lesser Antillean group that is nested within the Greater Antillean group and one to the Southern Lesser Antillean group that diverged from the Primary Mainland group early in the evolutionary history of *Anolis* (figure 1).

### (a) Bivariate allometry shows that species-specific differences in relative limb length are fully expressed at hatching

We first focused on the total fore-and hindlimb length (sum of all individual bones) in relation to body size to address when in ontogeny species-specific differences emerge. Species differences in relative limb length were substantial in both adults and hatchlings (all *p*-values < 0.001), and were detectable, albeit to a lesser degree, in embryos at stage 11, the earliest stage at which repeatable measurements were possible (forelimb: $F_{1,13} = 2.28$, $p = 0.025$; hindlimb: $F_{1,13} = 1.89$, $p = 0.064$; electronic supplementary material, table S2). Using mean values for each of the 15 species at hatching and as adults, these two life stages show a significant correlation of their relative hindlimb length (Pearson's product-moment correlation $r = 0.616$, *p*-value = 0.014) and a weak correlation of their relative forelimb length ($r = 0.361$, *p*-value = 0.187).

These results suggest that species differ both in the size of initial limb anlagen and in growth trajectories. Accordingly, an ANCOVA on total limb length revealed species differences in both intercept and bivariate allometric slopes (electronic supplementary material, table S3). For total fore- and hindlimb length, nearly all slopes were significantly larger than 1, indicating that limbs grow proportionally faster than the trunk (figure 2; electronic supplementary material, table S4). The largest species contrasts in growth rates (i.e. slopes) tended to involve species with exceptionally low allometric slopes (e.g. *A. equestris*, *A. rejectus*; electronic supplementary material,

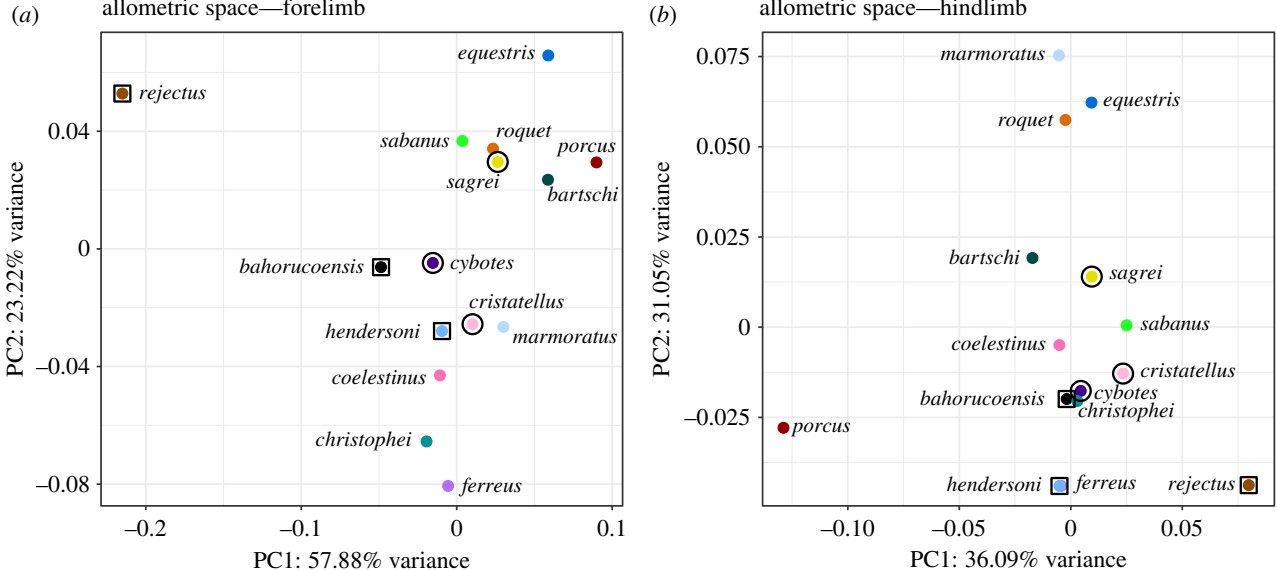

**Figure 3.** Allometric space of limb growth of six forelimb and seven hindlimb bones for 15 *Anolis* species. Allometric spaces of (*a*) fore- and (*b*) hindlimbs visualize the broad patterns of allometric growth of the 15 *Anolis* species. Allometric spaces were constructed from allometric vectors (PC1 coefficients) of each species. Species belonging to the same ecomorph as classified by Losos [23] are marked by a square (grass-bush) or a circle (trunk-ground). (Online version in colour.)

table S5). In agreement with these results on total limb length, MANCOVA on individual limb elements showed significant species and species-by-size effects (electronic supplementary material, table S3) which were equally distributed across individual limb elements (electronic supplementary material, table S6).

## (b) Multivariate ontogenetic allometries are conserved across *Anolis* species

While these results demonstrate that limbs of different *Anolis* species grow apart in terms of overall length relative to body size, they do not address if the relative proportions of individual limb elements are changing over ontogeny, and if this change is consistent across species. To address these questions, we first tested if the multivariate ontogenetic allometries differ between *Anolis* species.

Calculating the angles between allometric vectors for pairs of species revealed that the alignment between allometric trajectories generally lies within the 95% CIs of the bootstrapped angles, and are therefore consistent with one ontogenetic allometry shared by all 15 *Anolis* species (electronic supplementary material, tables S7 and S8; mean angle for forelimb was 6.03° and for hindlimb 4.10°). Out of 105 species pairs, only 14 (forelimb; 13.3%) and 11 (hindlimb; 10.5%) had angles between allometric trajectories that were significantly larger than those expected from a shared ontogenetic allometric (electronic supplementary material, tables S7 and S8).

The difference in multivariate allometry between species can be visualized using a second PCA on the allometric vectors of all 15 species. The result describes an allometric space that is commonly used to reveal clustering of species according to shared patterns of allometric growth [15,35]. Constructing such allometric spaces revealed that *A. rejectus* (forelimb and hindlimb) and *A. porcus* (hindlimb) are differentiated from all other species on PC1 of this allometric space (figure 2). The other species formed a continuous distribution along PC1 and PC2. It is noticeable that species belonging to the same ecomorph did not cluster tightly with each other and that

ecomorphs did not occupy a private location in allometric space (figure 3). This again illustrates how similar ontogenetic multivariate allometries are for these 15 species, despite the large differences in limb length relative to body size.

## (c) Major axis of evolutionary allometries are not well aligned with ontogenetic allometries

We first approached phylogenetic patterns of covariation in fore- and hindlimb elements through a phylogenetic PCA (pPCA) of all 267 *Anolis* species. This revealed no clear differentiation between the three major groups (Primary Mainland, Secondary Mainland and Antilles; electronic supplementary material, figure S3). The first principal component (pPC1) explained 93.64% (forelimb; using a standard PCA: 96.21%) and 95.01% (hindlimb; standard PCA: 96.13%) of the total variation and was tightly correlated with the centroid size of the pelvic girdle, a proxy for body size (forelimb: $r_P = 0.971$; hindlimb: $r_P = 0.951$; both $p$-value $< 0.001$; standard PCA: forelimb: $r_P = 0.971$; hindlimb: $r_P = 0.950$; both $p$-value $< 0.001$). We therefore interpret pPC1 as an allometric vector that describes how individual limb elements scale with overall limb length.

Before we addressed to what extent ontogenetic and evolutionary allometries are aligned with each other, we first asked whether patterns of evolutionary allometries are already established at the hatching stage. To this end, we compared the angles between evolutionary allometric vectors derived from hatchlings and adults of our 15 focal species. We found that the observed angles ($\Theta_{forelimb}$: 5.42°; $\Theta_{hindlimb}$: 4.97°; for results from an alternative analysis using standard PCA instead of phylogenetic PCA, see electronic supplementary material, table S5) lie within the 95% confidence interval of the bootstrap distribution derived from the sampling distribution of perfectly aligned allometric vectors (95% CI$_{forelimb}$: 8.19°; 95% CI$_{hindlimb}$: 9.28°; electronic supplementary material, table S9). This indicates that the evolutionary allometry of the 15 species is indistinguishable between hatchlings and adults.

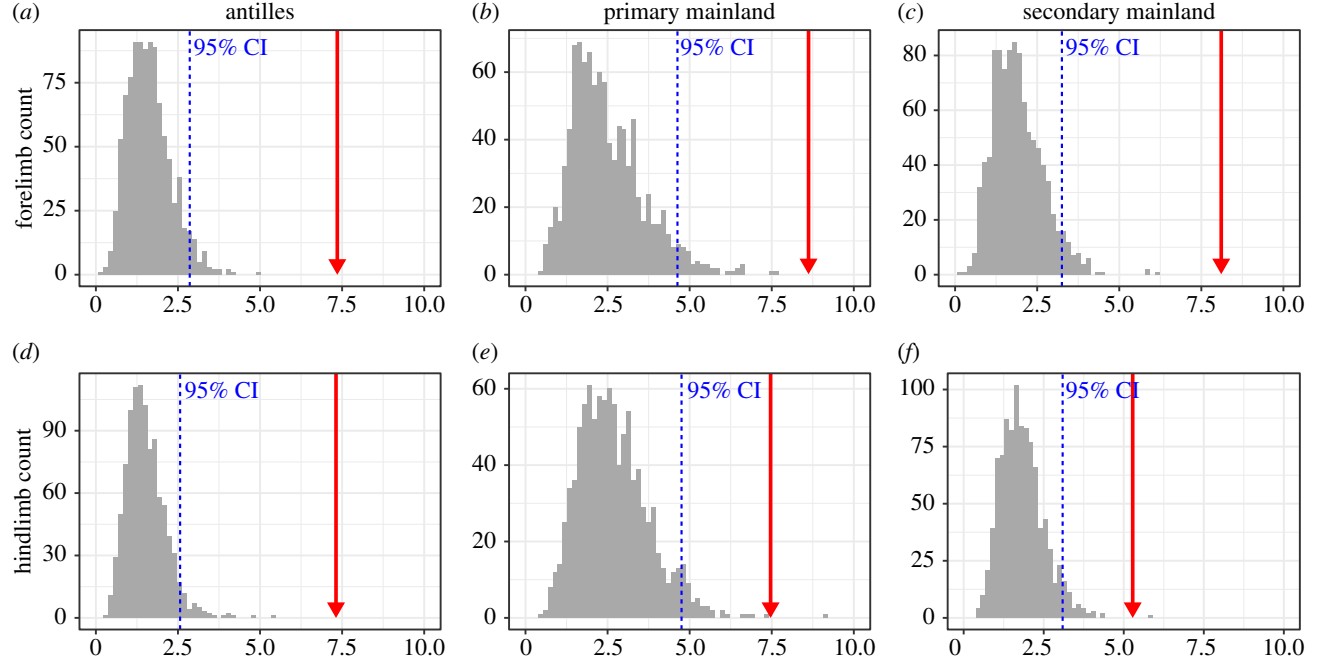

**Figure 4.** Alignment between ontogenetic allometry and phylogenetic allometry. Each plot represents a comparison between the ontogenetic allometric vector and the phylogenetic allometric vectors of the Antilles group (a,d), the Primary Mainland group (b,e) and the Secondary Mainland group (c,f) for fore- (a–c) and hindlimbs (d–f). The ontogenetic allometry is derived from species that belong to the Antillean group. Each plot shows the observed angles ($\Theta_{\text{ont-Ant}}$, $\Theta_{\text{ont-MLpri}}$, $\Theta_{\text{ont-MLsec}}$) between ontogenetic and phylogenetic allometric vectors as a red arrow. Grey bars indicate the frequency distribution of angles $\Theta$ drawn from modified datasets that simulated perfect alignment (derived from 1000 bootstrap replicates), and blue dashed lines mark their 95% confidence intervals. For results of an alternative analysis that used standard PCA instead of phylogenetic PCA, see electronic supplementary material, figure S4. (Online version in colour.)

To test if the major axis of ontogenetic growth of the limb bones is similar to the pattern of evolutionary diversification of limb morphology in the major groups, we assessed whether or not the coefficients of the allometric vectors showed similar patterns. Using bootstrapping of the ontogenetic major axis (common PC1 of 14 species since *A. roquet* was excluded from this analysis; see Material and methods), we found that the ontogenetic allometric vectors (coefficients of cPC1) of both fore- and hindlimbs deviated significantly from isometry (electronic supplementary material, tables S10 and S11). The ontogenetic coefficients for the four long bones (humerus, ulna, femur and tibia) were positive (i.e. a proportional increase relative to the other limb elements). By contrast, evolutionary coefficients for the long bones were nearly all significantly more negative. Similarly, the metatarsus showed a positive allometric growth pattern in ontogeny, but its evolutionary allometry revealed isometry in all three major groups. In general, the three major groups exhibited largely similar patterns of evolutionary allometry, evidenced by, for example, consistently positive coefficients of the last phalangeal element of the hindlimb. This was supported by quantifying the alignment between pairs of evolutionary allometric vectors that demonstrated that evolutionary allometries were highly similar between the three groups (all angles < 5.35°; electronic supplementary material, tables S12–15).

This qualitative pattern was quantitatively confirmed by assessing the alignment in terms of angles between the cPC1 vector of the ontogenetic dataset to the pPC1 vector of its native group (i.e. the Antillean group) and to the two more distantly related mainland groups. We found that the ontogenetic allometric vector (cPC1) of our 14 focal species was most closely aligned with the pPC1 of its native group for the forelimb ($\Theta_{\text{ont-Ant}}$: 7.27°; $\Theta_{\text{ont-MLpri}}$: 8.60°; $\Theta_{\text{ont-MLsec}}$: 8.10°), but not for the hindlimb ($\Theta_{\text{ont-Ant}}$: 7.29°; $\Theta_{\text{ont-MLpri}}$:

7.45°; $\Theta_{\text{ont-MLsec}}$: 5.24°). Comparing these angles against a bootstrapped distribution derived from a shared onto-phylogenetic allometric axis demonstrated that ontogenetic and phylogenetic axis are not well aligned with each other (figure 4; electronic supplementary material, figure S4).

## 4. Discussion

Unravelling the factors that shape adaptive evolution requires studies of both generative and selective processes. Decades of ecological research on *Anolis* lizards have come a long way in establishing the adaptive function of species differences in limb morphology, and many studies have demonstrated that traits such as relative limb length are under selection in contemporary populations [38–42]. By contrast, the developmental basis of variation in limb morphology and its consequences for the adaptive diversification of *Anolis* are little studied to date (but see [24,25]). Studies that investigate how morphological characters co-develop and coevolve are useful in this respect, and here we emphasize three aspects in particular.

Firstly, comparisons of growth trajectories between diverse morphologies can identify the developmental periods that are most evolutionarily labile, thereby pointing towards developmental mechanisms that are responsible for morphological evolvability. Our results demonstrate that, in *Anolis*, species differences in limb morphology are well established at hatching (both in terms of total relative length as well as the proportion of individual limb elements). This suggests that post-hatching regulation of limb growth might be a less important mechanism for the generation of adaptive variation than pre-hatching mechanisms. However, since our study did not include post-hatching stages, we cannot infer if growth trajectories in juveniles differ between *Anolis*

species. Nevertheless, this conclusion is consistent with a previous study [12], which showed that modification of the plastic responses to mechanical stress imposed by running versus climbing contributed little to adaptive evolution of the locomotor skeleton in *Anolis*. However, plasticity might still play a role in explaining differential bone growth in embryonic development since mechanical stress induced by motility of limbs *in ovo* is known to modulate growth rates of long bones (see e.g. [43]). The molecular mechanisms that promote limb bone growth include the paracrine signalling of the insulin-like growth factor (IGF) pathway [44,45] and of the bone morphogenetic protein (BMP) pathway [46] that stimulate chondrocyte proliferation and differentiation.

Irrespective of the mechanism underlying differential growth of limb element, our results demonstrate that differential growth alone is not responsible for the species differences in relative limb length in *Anolis* lizards, since these differences are to some extent manifested already in the early limb anlagen. Furthermore, we show that differential growth cannot account for species differences in the relative proportion of limb bones. Thus, the results from both bivariate and multivariate analysis of limb allometry suggest that the adaptive features that define *Anolis* habitat specialists are to a large extent established at the earliest phase of limb development, prior to or during the emergence of cartilaginous condensations. Further pinpointing the exact emergence of species-specific differences will require a modified methodology that does not rely on quantifying cartilaginous structures (e.g. [47]).

These results broadly agree with the findings of Wakasa *et al.* [25], who found that the difference in relative limb length between the short-limbed *A. angusticeps* and the long-limbed *A. sagrei* are evident before the formation of cartilaginous anlagen, and accumulate throughout embryonic and even post-hatching development. By contrast, another study by Sanger and colleagues [24] found that the long bones of two pairs of trunk-ground and trunk-crown specialists (ecomorphs that differ in relative limb length [23]) exhibit equal growth rates in their long bones relative to body size (i.e. bivariate allometric slopes [24]). While these observations agree with our conclusions in that species differences indeed are to a large extent established early in limb development (see also [47]), the lack of differential growth rates reported in this previous study [24] may be attributed to a more narrow taxon sampling that did not include more unique morphologies (e.g. our study included both ecomorph and non-ecomorph species). However, since in our study differential growth did not account for differences in the relative proportions of different limb bones, we suggest that much of the adaptive diversity in limb morphology in *Anolis* involves regulatory changes to genes involved in the early patterning of the limb, such as *HoxA10*, *HoxD11* and *HoxD12* [48]. Interestingly, the mechanisms of bone element morphogenesis differ between long bones, metapodials and phalangeal elements [49], which suggests that changes in limb length and proportions can result from a variety of different mechanisms, providing promising targets for future studies. Unfortunately, comparative developmental genetic data on the regulation of limb morphology are limited (but see [46]), but successful artificial selection for long-limbed mice largely occurred through an increase in the initial number of proliferative chondrocytes [50]. This is consistent with the early-acting mechanisms we predict to be responsible for the adaptive diversification of limb morphology in *Anolis*. Unravelling the developmental genetic basis of these mechanisms, together with genomic association studies, would be a powerful approach for identifying the developmental variabilities upon which selection can act, as exemplified by the research program on the beaks of Darwin's finches [51–53].

Since adaptive phenotypic change has to arise through developmental change, a second reason to study ontogenetic variability is that it can identify conserved developmental lines of least resistance that help to explain why adaptive diversification occurred in some directions and not others [4,54,55]. For example, jaws of cichlid fish show variational properties in development that appear to have directed evolutionary diversification [56,57]. While the ontogenetic allometry of the *Anolis* limb is highly conserved, we found that it was poorly aligned with the major axis of evolutionary allometry of both Antillean and mainland *Anolis*. In particular, the proportional increase in the length of long bones (proximal elements) was positive in embryonic development, but tended to be negative across species. Similarly, *Anolis* phalangeal (distal) elements exhibit patterns of coevolution with other limb bones that are distinct from their co-development. As explained above, these results suggest that the evolution of the *Anolis* limb involved modifications of developmental mechanisms early in ontogeny. These mechanisms may themselves exhibit developmental bias, which has been suggested to explain why the distal-most phalanges show the greatest variability and evolvability in birds [58]. Distal elements do appear to be particularly evolvable in *Anolis* as well. Across the phylogeny, differences in the length of distal bones account for much of the variation in relative limb length of different *Anolis* ecomorphs [25]. For example, Toro *et al.* [59] found that differences in the relative limb length between *A. sagrei* and *A. carolinensis* were most pronounced in the metatarsus and the longest toe. Similarly, Mahler *et al.* [60] reported that, among 22 morphological traits measured for 81 primarily Greater Antillean *Anolis* species, the length of the longest digit loaded highest on the first principal component. In addition to a possible developmental bias, the results for *Anolis* emphasize the functional importance of the phalangeal elements, which might be driven by their importance for the lizard's clinging ability, enabled through lamellae that cover the ventral side of phalangeal elements [61].

Finally, exploring the consistency of morphological covariation across different clades or environments can reveal insights into the relationship between adaptive diversification and developmental innovation [62]. For example, the butterfly genus *Heteropsis* has evolved correlations between eyespot colour patterns and elevated evolutionary rates that deviate from other Mycalesina butterflies, and this macroevolutionary shift coincides with a developmental innovation in eye spot development [63]. In *Anolis*, the major axis of evolutionary limb allometry has been conserved throughout the Antillean and two mainland radiations. Thus, despite Greater Antillean *Anolis* having generated ecomorph specialists that are (nearly [64]) absent from the mainland *Anolis* fauna (e.g. grass-bush, twig and crown-giant [65,66]), this adaptive radiation does not appear to involve profound changes to the underlying generative processes. Instead, the morphological variability of the *Anolis* limb seems perfectly capable of exploiting ecological opportunities, and the absence of certain morphologies from the mainland is better explained by ecological factors than by developmental innovations on islands [13,67]. This is consistent

with a macroevolutionary perspective that suggests that drastic modifications of limb proportions are rare. In mammals, such innovations are restricted to bats [46,62], jerboas [68] and arguably humans [69] (see also the conflicting conclusions regarding kangaroos and their allies; e.g. [70,71]). Squamate reptiles are understudied in this respect, although there are several recent studies on the evolutionary modularity of the skull that demonstrate the utility of this macroevolutionary approach [72,73]. Given that limb reduction is a rather frequent evolutionary event in squamates [74], it would be interesting to address the extent to which this is associated with changes in modularity and integration of different limb bones.

## 5. Conclusion

Our results show that both ontogenetic and phylogenetic allometries in limb bones are surprisingly conserved across *Anolis* species. However, evolutionary diversification of limb morphology has not followed the ontogenetic growth trajectory of limb bones, but is the result of modifications of developmental processes that act during early developmental stages.

Ethics. All work was conducted according to the Lund University Local Ethical Review Process under the permit number Dnr M 31-16.

Data accessibility. Raw CT scans of adult *Anolis* specimens are deposited at MorphoSource under the project ID P1059, title 'Anolis sp.'. Code and datasets, including a list of DOIs of individual CT scans deposited at MorphoSource, are available from the Dryad digital Repository: https://doi.org/10.5061/dryad.jsxksn08d [75].

Authors' contributions. N.F.: conceptualization, data curation, formal analysis, funding acquisition, investigation, methodology, project administration, resources, supervision, visualization, writing-original draft, writing-review and editing; I.S.C.J.: data curation, investigation, methodology, resources, writing-review and editing; E.V.d.C.: data curation, formal analysis, methodology, writing-review and editing; T.U.: conceptualization, formal analysis, funding acquisition, investigation, methodology, project administration, resources, writing-original draft, writing-review and editing

All authors gave final approval for publication and agreed to be held accountable for the work performed therein.

Competing interests. We declare we have no competing interests.

Funding. This research was supported by a grant from the John Templeton Foundation (60501) to T.U., a grant from the Royal Physiographic Society of Lund to N.F., a Wenner-Gren postdoctoral fellowship to N.F., and a Wallenberg Academy Fellowship from the Knut and Alice Wallenberg Foundation to T.U.

Acknowledgements. We are grateful to private breeders for supplying us with stock animals. We thank David Blackburn and Edward L Stanley for providing us with access to the Nanoscale Research Facility (University of Florida, US) where all micro-CT scanning was conducted. We thank Dan Warner and Tim Mitchell for assistance with collecting animals in the field.

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
