## [Peer Review File · Proceedings of the Royal Society B: Biological Sciences]

Review History

RSPB-2021-0226.R0 (Original submission)

Review form: Reviewer 1

Recommendation

Accept with minor revision (please list in comments)

Scientific importance: Is the manuscript an original and important contribution to its field?

Good

General interest: Is the paper of sufficient general interest?

Excellent

Quality of the paper: Is the overall quality of the paper suitable?

Good

Is the length of the paper justified?

Yes

Should the paper be seen by a specialist statistical reviewer?

No

Do you have any concerns about statistical analyses in this paper? If so, please specify them explicitly in your report.

No

It is a condition of publication that authors make their supporting data, code and materials available - either as supplementary material or hosted in an external repository. Please rate, if applicable, the supporting data on the following criteria.

Is it accessible?

N/A

Is it clear?

N/A

Is it adequate?

N/A

Do you have any ethical concerns with this paper?

No

Comments to the Author

This is a very nice paper that compares embryonic allometry in limb length in 15 anole species with patterns exhibited across the entire clade. The results show that patterns of limb growth are strongly conserved and, most importantly, that interspecific differences among adults are fundamentally different from the conserved patterns of growth displayed within species.

This is a well-done and very labor-intensive study that presents new and important data. I only have two reservations.

First, I feel that the authors could make the key finding of this study even more clear: conserved growth trajectories do not constrain evolutionary diversification. This paper refutes the idea that generative processes constrain the realm of the possible and thus limit the impact of selection. This point needs to be said more clearly and strongly: in the debate of the constraints versus selectionist schools, this paper strongly supports the primacy of natural selection.

Along these lines, the authors should look into the work of Russell Powell, who has staked out a pretty hard-core perspective in his 2012 paper and his 2015 paper with Mariscal (<http://dx.doi.org/10.1098/rsfs.2015.0040>). He also has recently published a book which I suspect elaborates further on his arguments for the role of constraint in shaping anole evolution. The data presented here are relevant to Powell's arguments.

Second, I have to comment on the authors' uncharitable treatment of the 2012 paper by Sanger et al. which also appeared in the Proceedings. That paper is the intellectual forebear of this one and deserves much stronger recognition and acknowledgment. As it is, the authors mention it only in ways that minimize what it showed, rather than clearly stating how that paper laid the groundwork for this one. That's not to say that this present paper is derivative – it goes substantially farther in multiple respects. It doesn't diminish what the authors have accomplished to fairly represent the intellectual history of investigation of this topic.

Here are some minor comments:

42: I haven't gone back to check, but my recollection is that there were actually many such studies in the heyday of allometry research. The idea of comparing evolutionary and ontogenetic allometry was conceptualized then.

54: the “ecomorph concept” is not well articulated and won’t be understood by readers not already familiar with the system

98-99: a minor point, but incubation time undoubtedly is affected by temperature, and because species have different thermal physiologies, norms of reaction are likely to vary across species. Hence, measuring incubation time at one temperature thus provides limited information. Also, the chosen temperature is not justified.

144-146: why are different proxies for body size used for embryos and hatchlings vs. adults?

177: unclear what is meant by “unique morphologies”

181-185: it might be more straightforward to briefly describe anole phylogeny: anoles colonized the Caribbean twice from the mainland, once giving rise to the species in the Greater and northern Lesser Antilles, the second time giving rise to the clade in the southern Lesser Antilles

248-252: Bonferroni correction is overly conservative, as illustrated by the results reported here. With 105 tests, one would expect 5.25 false positives, yet Bonferroni correction disqualifies 11 and 15 positive results. This is an excessive loss in power. I would either delete the Bonferroni correction or, minimally, point out the tradeoff between Type I and Type II error.

327-329: this statement creates a straw-man of what the authors label the “plasticity first” hypothesis. That hypothesis never claimed that the major differences in limb length among ecomorphs is the result of plasticity, nor, contrary to the authors’ recent work, did it suggest that plasticity played a major role in ecomorph differentiation. The results reported here do not refute the possibility that plasticity can alter limb length to some extent, as the authors themselves showed in their previous work. Moreover, a large body of research, some of it cited in this paper, indicates the adaptive value of minor changes in limb length; given this, it is an over-reach to conclude that plasticity is a “rather unimportant mechanism for the generation of adaptive variation.”

355: the most “extreme” morphologies are not necessarily exhibited by “non-ecomorph” species. Incidentally, some have criticized the term “non-ecomorph” as defining species by what they aren’t; for that reason, “unique” has been used more recently, which also emphasizes the evolutionary convergence aspect of the ecomorphs, a point that could have been made clearer when “ecomorphs” were introduced

393-394: subject-verb tense disagreement

Review form: Reviewer 2 (Kathryn Kavanagh)

Recommendation

Major revision is needed (please make suggestions in comments)

Scientific importance: Is the manuscript an original and important contribution to its field?

Good

General interest: Is the paper of sufficient general interest?

Good

Quality of the paper: Is the overall quality of the paper suitable?

Good

Is the length of the paper justified?

Yes

Should the paper be seen by a specialist statistical reviewer?

Yes

Do you have any concerns about statistical analyses in this paper? If so, please specify them explicitly in your report.

No

It is a condition of publication that authors make their supporting data, code and materials available - either as supplementary material or hosted in an external repository. Please rate, if applicable, the supporting data on the following criteria.

Is it accessible?

Yes

Is it clear?

No

Is it adequate?

No

Do you have any ethical concerns with this paper?

No

Comments to the Author

Review of RSPB-2021-0226 Feiner

Feiner et al. measured proportions in limb bones of 15 species of anoles from hatching to adulthood, and from adult anoles from another 267 species. They compared developmental allometric growth and evolutionary allometric growth in these datasets to see if they are related. If the allometries were 'aligned' they propose that would indicate early developmental integration channels diversification in the clade.

Unfortunately, they did not measure the stage where adjacent elements were developmentally integrated and influencing proportions, so the question remains unanswered at that level.

They wanted to determine when differences in proportions arise, with the stated goals...

"when in ontogeny do species specific relative limb length is established"

"do the different proportions emerge through differential rate of growth of individual bones or different size of limb anlagen?"

However they are not able to answer these questions because they do not measure anlagen at the time of formation; they only measure later development once all the elements are formed. Limb and digit elements form in sequence. The embryos were not evaluated prior to stage 11. By this time, the final phalanx is formed and the interdigital tissue is regressing, and (they showed) the final proportions are already established. So, as they discovered, this paper is actually comparing postmorphogenetic growth allometries with evolutionary patterns of proportion changes.

The authors did answer the question of whether embryonic postmorphogenetic (aka fetal) allometries match evolutionary allometries - and the answer is that they did not. This makes sense, and it would be useful to expand more on this explanation in this large morphologically diverse sample of anoles. These kinds of datasets are extremely rare and laborious to obtain, so this would be an excellent addition to the literature on allometry and growth. For example, their mention of the lack of support for "plasticity-led" evolution is also useful in this context. There

may be other reasons to consider why adult proportions would be stabilized so early in development.

Earlier in development, the distal phalanges will be still forming while the proximal elements will be fully formed and growing at a slower rate to 'arrive' at the final proportions just after the tip forms.

If the authors wish to understand how limb and digit proportions arise and are developmentally regulated, they should explore the literature on joint patterning. In the digits, the individual skeletal elements are not formed as individual condensations, but rather the joint positions form by downregulating the cartilage pathway within the condensation, creating a particular joint position. It is at that time when the proportions are established. (e.g. Kavanagh et al., 2013 PNAS, Supplementary 2A). The allometries in this brief stage are where developmental regulation of proportion occurs.

To make the data more accessible to researchers studying phenotypes, I wish more intuitive figures were prepared for the main manuscript including explanations that refer back to phenotypes and individual proportions to help with understanding. The supplementary helps a lot though. A figure with images of the cleared-and-stained limbs in the developmental series would be a great addition.

"343 both bivariate and multivariate analysis of limb allometry suggest that the adaptive features 344 that define Anolis habitat specialists are to a large extent established at the earliest phase of 345 limb development, prior to the emergence of cartilaginous condensations."
 – Rephrase to "...prior to or during the emergence..."

Decision letter (RSPB-2021-0226.R0)

23-Mar-2021

Dear Dr Feiner:

Your manuscript has now been peer reviewed and the reviews have been assessed by an Associate Editor. The reviewers' comments (not including confidential comments to the Editor) and the comments from the Associate Editor are included at the end of this email for your reference. As you will see, the reviewers and the Editors have raised some concerns with your manuscript and we would like to invite you to revise your manuscript to address them.#

Both reviewers request some substantial revisions to the text. Further analyses may not be necessary-- you can consider and address this in your Response. However, it will be vital to give fair credit to Sanger et al. and other studies.

Furthermore, it was noted in review that "Images are unlabeled unspecified elements in the MorphoSource Anolis sp. so impossible to review."-- please amend this and provide DOIs for these data with the revised MS.

To submit your revision please log into <http://mc.manuscriptcentral.com/prsb> and enter your Author Centre, where you will find your manuscript title listed under "Manuscripts with

Decisions." Under "Actions", click on "Create a Revision". Your manuscript number has been appended to denote a revision.

Research ethics:

Use of animals and field studies:

It is a condition of publication that you make available the data and research materials supporting the results in the article. Please see our Data Sharing Policies (<https://royalsociety.org/journals/authors/author-guidelines/#data>). Datasets should be deposited in an appropriate publicly available repository and details of the associated accession number, link or DOI to the datasets must be included in the Data Accessibility section of the article (<https://royalsociety.org/journals/ethics-policies/data-sharing-mining/>). Reference(s) to datasets should also be included in the reference list of the article with DOIs (where available).

All supplementary materials accompanying an accepted article will be treated as in their final form. They will be published alongside the paper on the journal website and posted on the online figshare repository. Files on figshare will be made available approximately one week before the

accompanying article so that the supplementary material can be attributed a unique DOI. Please try to submit all supplementary material as a single file.

Please submit a copy of your revised paper within three weeks. If we do not hear from you within this time your manuscript will be rejected. If you are unable to meet this deadline please let us know as soon as possible, as we may be able to grant a short extension.

Best wishes,
Dr John Hutchinson, Editor
mailto:proceedingsb@royalsociety.org

Associate Editor

Board Member: 1

Comments to Author:

The reviewers agree that the study has considerable merit and would be of interest to the readers of Proceedings B. They raise various concerns about the context and interpretation of the data. Most importantly, concern was expressed about a mismatch between the ontogenetic stages studied and the interpretations offered, particularly the discussion of the relative sizes of limb anlagen. This must be either clarified in the text or addressed through additional analysis. In addition, the existing literature on the subject should be fully acknowledged, including the contributions of others as well as the knowledge gaps that this manuscript seeks to fill, and (as appropriate) whether your results agree with previous studies.

Reviewer(s)' Comments to Author:

Referee: 1

Comments to the Author(s)

This is a very nice paper that compares embryonic allometry in limb length in 15 anole species with patterns exhibited across the entire clade. The results show that patterns of limb growth are strongly conserved and, most importantly, that interspecific differences among adults are fundamentally different from the conserved patterns of growth displayed within species.

This is a well-done and very labor-intensive study that presents new and important data. I only have two reservations.

First, I feel that the authors could make the key finding of this study even more clear: conserved growth trajectories do not constrain evolutionary diversification. This paper refutes the idea that generative processes constrain the realm of the possible and thus limit the impact of selection. This point needs to be said more clearly and strongly: in the debate of the constraints versus selectionist schools, this paper strongly supports the primacy of natural selection.

Along these lines, the authors should look into the work of Russell Powell, who has staked out a pretty hard-core perspective in his 2012 paper and his 2015 paper with Mariscal (<http://dx.doi.org/10.1098/rsfs.2015.0040>). He also has recently published a book which I suspect elaborates further on his arguments for the role of constraint in shaping anole evolution. The data presented here are relevant to Powell's arguments.

Second, I have to comment on the authors' uncharitable treatment of the 2012 paper by Sanger et al. which also appeared in the Proceedings. That paper is the intellectual forebear of this one and deserves much stronger recognition and acknowledgment. As it is, the authors mention it only in ways that minimize what it showed, rather than clearly stating how that paper laid the groundwork for this one. That's not to say that this present paper is derivative—it goes substantially farther in multiple respects. It doesn't diminish what the authors have accomplished to fairly represent the intellectual history of investigation of this topic.

Here are some minor comments:

42: I haven't gone back to check, but my recollection is that there were actually many such studies in the heyday of allometry research. The idea of comparing evolutionary and ontogenetic allometry was conceptualized then.

54: the "ecomorph concept" is not well articulated and won't be understood by readers not already familiar with the system

98-99: a minor point, but incubation time undoubtedly is affected by temperature, and because species have different thermal physiologies, norms of reaction are likely to vary across species. Hence, measuring incubation time at one temperature thus provides limited information. Also, the chosen temperature is not justified.

144-146: why are different proxies for body size used for embryos and hatchlings vs. adults?

177: unclear what is meant by "unique morphologies"

181-185: it might be more straightforward to briefly describe anole phylogeny: anoles colonized the Caribbean twice from the mainland, once giving rise to the species in the Greater and northern Lesser Antilles, the second time giving rise to the clade in the southern Lesser Antilles

248-252: Bonferroni correction is overly conservative, as illustrated by the results reported here. With 105 tests, one would expect 5.25 false positives, yet Bonferroni correction disqualifies 11 and 15 positive results. This is an excessive loss in power. I would either delete the Bonferroni correction or, minimally, point out the tradeoff between Type I and Type II error.

327-329: this statement creates a straw-man of what the authors label the "plasticity first" hypothesis. That hypothesis never claimed that the major differences in limb length among ecomorphs is the result of plasticity, nor, contrary to the authors' recent work, did it suggest that plasticity played a major role in ecomorph differentiation. The results reported here do not refute the possibility that plasticity can alter limb length to some extent, as the authors themselves showed in their previous work. Moreover, a large body of research, some of it cited in this paper, indicates the adaptive value of minor changes in limb length; given this, it is an over-reach to conclude that plasticity is a "rather unimportant mechanism for the generation of adaptive variation."

355: the most "extreme" morphologies are not necessarily exhibited by "non-ecomorph" species. Incidentally, some have criticized the term "non-ecomorph" as defining species by what they aren't; for that reason, "unique" has been used more recently, which also emphasizes the evolutionary convergence aspect of the ecomorphs, a point that could have been made clearer when "ecomorphs" were introduced

393-394: subject-verb tense disagreement

Referee: 2
 Comments to the Author(s)
 Review of RSPB-2021-0226 Feiner

Feiner et al. measured proportions in limb bones of 15 species of anoles from hatching to adulthood, and from adult anoles from another 267 species. They compared developmental allometric growth and evolutionary allometric growth in these datasets to see if they are related. If the allometries were 'aligned' they propose that would indicate early developmental integration channels diversification in the clade.

Unfortunately, they did not measure the stage where adjacent elements were developmentally integrated and influencing proportions, so the question remains unanswered at that level.

They wanted to determine when differences in proportions arise, with the stated goals...
 "when in ontogeny do species specific relative limb length is established"
 "do the different proportions emerge through differential rate of growth of individual bones or different size of limb anlagen?"

However they are not able to answer these questions because they do not measure anlagen at the time of formation; they only measure later development once all the elements are formed. Limb and digit elements form in sequence. The embryos were not evaluated prior to stage 11. By this time, the final phalanx is formed and the interdigital tissue is regressing, and (they showed) the final proportions are already established. So, as they discovered, this paper is actually comparing postmorphogenetic growth allometries with evolutionary patterns of proportion changes.

The authors did answer the question of whether embryonic postmorphogenetic (aka fetal) allometries match evolutionary allometries - and the answer is that they did not. This makes sense, and it would be useful to expand more on this explanation in this large morphologically diverse sample of anoles. These kinds of datasets are extremely rare and laborious to obtain, so this would be an excellent addition to the literature on allometry and growth. For example, their mention of the lack of support for "plasticity-led" evolution is also useful in this context. There may be other reasons to consider why adult proportions would be stabilized so early in development.

Earlier in development, the distal phalanges will be still forming while the proximal elements will be fully formed and growing at a slower rate to 'arrive' at the final proportions just after the tip forms.

If the authors wish to understand how limb and digit proportions arise and are developmentally regulated, they should explore the literature on joint patterning. In the digits, the individual skeletal elements are not formed as individual condensations, but rather the joint positions form by downregulating the cartilage pathway within the condensation, creating a particular joint position. It is at that time when the proportions are established. (e.g. Kavanagh et al., 2013 PNAS, Supplementary 2A). The allometries in this brief stage are where developmental regulation of proportion occurs.

To make the data more accessible to researchers studying phenotypes, I wish more intuitive figures were prepared for the main manuscript including explanations that refer back to phenotypes and individual proportions to help with understanding. The supplementary helps a lot though. A figure with images of the cleared-and-stained limbs in the developmental series would be a great addition.

"343 both bivariate and multivariate analysis of limb allometry suggest that the adaptive features
 344 that define Anolis habitat specialists are to a large extent established at the earliest phase of
 345 limb development, prior to the emergence of cartilaginous condensations."
 – Rephrase to "...prior to or during the emergence..."

Author's Response to Decision Letter for (RSPB-2021-0226.R0)

See Appendix A.

RSPB-2021-0226.R1 (Revision)

Review form: Reviewer 1

Recommendation

Accept with minor revision (please list in comments)

Scientific importance: Is the manuscript an original and important contribution to its field?

Good

General interest: Is the paper of sufficient general interest?

Good

Quality of the paper: Is the overall quality of the paper suitable?

Good

Is the length of the paper justified?

Yes

Should the paper be seen by a specialist statistical reviewer?

No

Do you have any concerns about statistical analyses in this paper? If so, please specify them explicitly in your report.

No

It is a condition of publication that authors make their supporting data, code and materials available - either as supplementary material or hosted in an external repository. Please rate, if applicable, the supporting data on the following criteria.

Is it accessible?

Yes

Is it clear?

Yes

Is it adequate?

Yes

Do you have any ethical concerns with this paper?

No

Comments to the Author

The authors have done a nice job of revising their manuscript, which was already quite good. I only have two comments:

376-382: it may be relevant to note that angusticeps and sagrei are distantly related, whereas by design, Sanger et al. compared pairs of closely related species that differed in limb length.

402, paragraph: I continue to believe that the authors are failing to own up to the results of their study. The finding that evolutionary allometry fails to mirror ontogenetic allometry – indeed, is in the opposite direction – suggests that, in fact, development is not constraining evolution, that evolution is not occurring along lines of least resistance. The authors may have set out to find support for this hypothesis, but I feel they are not being upfront in acknowledging that the results would seem to falsify that hypothesis.

If nothing else, the authors should more explicitly discuss how these results sit in the context of their initial hypotheses.

Along those lines, the back half of the paragraph, starting on line 410 “this is consistent with...” does not make sense to me. I do not see how that material has any bearing on the preceding material, much less being consistent with it. On re-reading a third time, I guess the point has to do with “distal” vs. “long” bones. The authors should make this more explicit for those who are not anatomists. Regardless, since the contrasting pattern of positive versus negative allometry pertains to the long bones, the material on distal points seems tangential.

Review form: Reviewer 2

Recommendation

Accept as is

Scientific importance: Is the manuscript an original and important contribution to its field?

Excellent

General interest: Is the paper of sufficient general interest?

Excellent

Quality of the paper: Is the overall quality of the paper suitable?

Excellent

Is the length of the paper justified?

Yes

Should the paper be seen by a specialist statistical reviewer?

No

Do you have any concerns about statistical analyses in this paper? If so, please specify them explicitly in your report.

No

It is a condition of publication that authors make their supporting data, code and materials available - either as supplementary material or hosted in an external repository. Please rate, if applicable, the supporting data on the following criteria.

Is it accessible?

Yes

Is it clear?

Yes

Is it adequate?

N/A

Do you have any ethical concerns with this paper?

No

Comments to the Author

The revisions are sufficient.

Decision letter (RSPB-2021-0226.R1)

28-May-2021

Dear Dr Feiner

I am pleased to inform you that your manuscript RSPB-2021-0226.R1 entitled "A highly conserved ontogenetic limb allometry and its evolutionary significance in the adaptive radiation of *Anolis* lizards" has been accepted for publication in *Proceedings B*. Congratulations!!

The referee(s) have recommended publication, but also suggest some minor revisions to your manuscript. Therefore, I invite you to respond to the referee(s)' comments and revise your manuscript. Because the schedule for publication is very tight, it is a condition of publication that you submit the revised version of your manuscript within 7 days. If you do not think you will be able to meet this date please let us know.

[http://datadryad.org/submit?journalID=RSPB&manu=\(Document not available\)](http://datadryad.org/submit?journalID=RSPB&manu=(Document not available)) which will take you to your unique entry in the Dryad repository. If you have already submitted your data to dryad you can make any necessary revisions to your dataset by following the above link.

Please see <https://royalsociety.org/journals/ethics-policies/data-sharing-mining/> for more details.

Sincerely,

Dr John Hutchinson

Associate Editor:

Board Member: 1

Comments to Author:

The reviewers agree that the revised version is very good, and only a few very minor points remain to be addressed. Most importantly, please indicate somewhere (in the supplementary info. section of the manuscript and/or the project description on MorphoSource) that a list of DOI's is available and where it is located. Currently there's no indication that the DOI's are on Dryad. In addition, you may wish to clarify anatomical terms (e.g., "long bones") as suggested by Reviewer 1.

Reviewer(s)' Comments to Author:

Referee: 1

Comments to the Author(s)

The authors have done a nice job of revising their manuscript, which was already quite good. I only have two comments:

376-382: it may be relevant to note that *angusticeps* and *sagrei* are distantly related, whereas by design, Sanger et al. compared pairs of closely related species that differed in limb length.

402, paragraph: I continue to believe that the authors are failing to own up to the results of their study. The finding that evolutionary allometry fails to mirror ontogenetic allometry – indeed, is in the opposite direction – suggests that, in fact, development is not constraining evolution, that evolution is not occurring along lines of least resistance. The authors may have set out to find support for this hypothesis, but I feel they are not being upfront in acknowledging that the results would seem to falsify that hypothesis.

If nothing else, the authors should more explicitly discuss how these results sit in the context of their initial hypotheses.

Along those lines, the back half of the paragraph, starting on line 410 “this is consistent with...” does not make sense to me. I do not see how that material has any bearing on the preceding material, much less being consistent with it. On re-reading a third time, I guess the point has to do with “distal” vs. “long” bones. The authors should make this more explicit for those who are not anatomists. Regardless, since the contrasting pattern of positive versus negative allometry pertains to the long bones, the material on distal points seems tangential.

Referee: 2

Comments to the Author(s)

The revisions are sufficient.

Author's Response to Decision Letter for (RSPB-2021-0226.R1)

See Appendix B.

Decision letter (RSPB-2021-0226.R2)

01-Jun-2021

Dear Dr Feiner

I am pleased to inform you that your manuscript entitled "A highly conserved ontogenetic limb allometry and its evolutionary significance in the adaptive radiation of *Anolis* lizards" has been accepted for publication in Proceedings B.

Data Accessibility section

Open Access

Paper charges

Sincerely,

Appendix A

Below is the report of our manuscript revision. We show sentences taken from the manuscript in italic, and parts newly inserted into the manuscript as underlined. Page and line numbers refer to positions in the document with tracked changes.

REVIEWER COMMENTS

Editor (Comments to Author):

Both reviewers request some substantial revisions to the text. Further analyses may not be necessary-- you can consider and address this in your Response. However, it will be vital to give fair credit to Sanger et al. and other studies.

Response: We have substantially revised our manuscript to better cover the existing body of literature. Please see our detailed explanations below.

Furthermore, it was noted in review that "Images are unlabeled unspecified elements in the MorphoSource Anolis sp. so impossible to review."-- please amend this and provide DOIs for these data with the revised MS.

Response: Thank you for picking this up. We have now included the file 'MuseumIndiv_DOIs' in our Dryad dataset which provides the individual DOI numbers for each of the 693 individual CT-scans of the adult individuals (available on MorphoSource) included in this study.

Associate Editor (Comments to Author):

The reviewers agree that the study has considerable merit and would be of interest to the readers of Proceedings B. They raise various concerns about the context and interpretation of the data. Most importantly, concern was expressed about a mismatch between the ontogenetic stages studied and the interpretations offered, particularly the discussion of the relative sizes of limb anlagen. This must be either clarified in the text or addressed through additional analysis. In addition, the existing literature on the subject should be fully acknowledged, including the contributions of others as well as the knowledge gaps that this manuscript seeks to fill, and (as appropriate) whether your results agree with previous studies.

Response: We thank the Associate Editor for the overall positive evaluation and for the succinct summary of the reviewers' comments. We have taken this opportunity to include a more thorough treatment of the existing literature into our revised manuscript (expanded Introduction and Discussion). We also emphasized limitations of our study, namely the focus on pre-hatching stages in the ontogenetic dataset. We believe that our modified manuscript has substantially improved and provides a better understanding on how our findings advance this field.

Reviewer #1 (Comments to Author):

This is a very nice paper that compares embryonic allometry in limb length in 15 anole species with patterns exhibited across the entire clade. The results show that patterns of limb growth are strongly conserved and, most importantly, that interspecific differences among adults are fundamentally different from the conserved patterns of growth displayed within species. This is a well-done and very labor-intensive study that presents new and important data. I only have two reservations.

Response: We thank the reviewer for the appreciation of our study and for the useful feedback.

First, I feel that the authors could make the key finding of this study even more clear: conserved growth trajectories do not constrain evolutionary diversification. This paper refutes the idea that generative processes constrain the realm of the possible and thus limit the impact of selection. This point needs to be said more clearly and strongly: in the debate of the constraints versus selectionist schools, this paper strongly supports the primacy of natural selection.

Response: We agree with the reviewer that in a polarized constraint-vs-selection debate, our findings could be interpreted as supporting a selectionist view. However, we wish to not re-enforce this dichotomous thinking since we do not see this as a fruitful debate. That is, selection and generative processes are not alternative forces, but rather interact in shaping phenotype distributions. The fact that we do not find an alignment between ontogenetic and evolutionary allometries does not mean that selection can produce any morphology regardless of the underlying developmental mechanisms. Developmental systems will always bias and co-determine patterns of diversifications, and the question is the extent to which these biases are persistent enough to exercise long-lasting effects on evolution. Since we believe the constraint vs selection debate rests on a false dichotomy we chose to not present this line of thought in our manuscript.

Along these lines, the authors should look into the work of Russell Powell, who has staked out a pretty hard-core perspective in his 2012 paper and his 2015 paper with Mariscal (<http://dx.doi.org/10.1098/rsfs.2015.0040>). He also has recently published a book which I suspect elaborates further on his arguments for the role of constraint in shaping anole evolution. The data presented here are relevant to Powell's arguments.

Response: As discussed above, we do not interpret our findings as 'refuting the idea that generative processes constrain (...) selection' (quote Reviewer), and therefore do not wish to expand on this aspect in our manuscript.

Second, I have to comment on the authors' uncharitable treatment of the 2012 paper by Sanger et al. which also appeared in the Proceedings. That paper is the intellectual forebear of this one and deserves much stronger recognition and acknowledgment. As it is, the authors mention it only in ways that minimize what it showed, rather than clearly stating how that paper laid the groundwork for this one. That's not to say that this present paper is derivative—it goes substantially farther in multiple respects. It

doesn't diminish what the authors have accomplished to fairly represent the intellectual history of investigation of this topic.

Response: Diminishing the importance of the Sanger et al. (2012, *Proc B*) paper was very far from our intention. We apologize if our manuscript gave this impression. In the revised version, we introduce Sanger *et al.*'s and also Wakasa *et al.*'s key findings already in the Introduction in order to give full credit to the work that we are building upon. We have also added references to a study by Andrews & Skewes to our Discussion which is relevant in this respect.

Original: *While limb morphology is arguably the most defining ecological feature of Anolis, the developmental basis of adaptive variation is poorly understood (but see [25, 26]).*

Revised (lines 62-70): *While limb morphology is arguably the most defining ecological feature of Anolis, the developmental basis of adaptive variation has received little attention. One study comparing limb ontogenetic growth between pairs of Anolis species belonging to different ecomorphs found that relative limb length differences are established very early in development and that differential growth contributes little, if anything, to ecomorph-specific differences [24]. Another study on two Anolis species confirmed the early appearance of species-specific limb length differences, but found that these increase over developmental time, even after hatching [25].*

Sentence added (lines 373-375): *Further pinpointing the exact emergence of species-specific differences will require a modified methodology that does not rely on quantifying cartilaginous structures (e.g., [49]).*

Sentence modified (lines 383, 384): *While these observations agree with our conclusions in that species differences indeed are to a large extent established early in limb development (see also [49]), (...)*

Reference added: *49. Andrews RM, Skewes SA. 2017 Developmental origin of limb size variation in lizards. *Evol. Dev.* 19, 136-146. (doi:10.1111/ede.12221).*

Here are some minor comments:

42: I haven't gone back to check, but my recollection is that there were actually many such studies in the heyday of allometry research. The idea of comparing evolutionary and ontogenetic allometry was conceptualized then.

Response: We agree with the reviewer that studies comparing evolutionary and ontogenetic allometries are rather common, but our intention here was to specifically point out the lack of such studies in the context of adaptive radiations. We have rephrased the sentence to make this point more clear.

Original: *While an increasing number of studies compare trait covariation across species [e.g., 1, 11-13], few studies of adaptive radiations have investigated ontogenetic trajectories [6, 14, 15].*

Revised (lines 43-46): *While an increasing number of studies compares trait covariation across species [e.g., 1, 11-14], and sometimes relate these to ontogenetic trajectories [6, 15, 16], little has been done in the context of adaptive radiations.*

54: the “ecomorph concept” is not well articulated and won’t be understood by readers not already familiar with the system

Response: We agree and have added more explanations to the Introduction. However, we would like to emphasize that we deliberately do not want to focus the attention of the reader on ecomorphs since our comparison includes many non-ecomorph species.

Original: *This match is captured by the ecomorph classification, which is commonly used to structure the phenotypic variation on the Greater Antillean islands [21-23].*

Revised (lines 57-62): *This match is captured by the ecomorph classification, which recognized up to six habitat specialist types (ecomorphs) that have repeatedly and convergently evolved on the four Greater Antillean islands [21-23]. However, not all Greater Antillean species are classified as ecomorphs and roughly 10% exhibit more unique morphologies.*

98-99: a minor point, but incubation time undoubtedly is affected by temperature, and because species have different thermal physiologies, norms of reaction are likely to vary across species. Hence, measuring incubation time at one temperature thus provides limited information. Also, the chosen temperature is not justified.

Response: It is correct that different *Anolis* species likely have different optimal incubation temperatures, but 26°C is a good compromise and lies within the acceptable range of all species included in our study. This is evidenced by the literature (the book ‘*Anolis*’ by Fläschendräger & Wijffels) and we have confirmed this by obtaining consistently high hatching success approaching 100% for all species included in this study. We have added this justification to the Methods section.

Sentence added (lines 112-114): *While different *Anolis* species likely have different optimal incubation temperatures [26], 26°C is a good compromise for the species included in the present work.*

Reference added: *26. Fläschendräger A, Wijffels L. 2009 *Anolis*. 2nd edition ed, NTV Natur und Tier-Verlag.*

In addition, we would like to re-emphasize that developmental time was not used as a variable in our analyses, but merely to obtain an overview of ontogenetic development (see figure S1).

Original: *Subsequently, embryos were dissected at 12 evenly spaced time points to obtain an ontogenetic series of each species that provides a broad overview of the variation of morphogenesis (electronic supplementary material, figure S1).*

Revised (lines 116-119): *On the basis of this species-specific incubation time, embryos were dissected at 12 evenly spaced time points to obtain an ontogenetic series of each species that provides a broad overview of the variation of morphogenesis (electronic supplementary material, figure S1).*

144-146: why are different proxies for body size used for embryos and hatchlings vs. adults?

Response: The reason for this is purely technical; Snout-vent-length (svl) is the standard measure of body size in herpetology, but this measure could not be reliably obtained for preserved

museum samples (body postures are usually strongly bent). We therefore used the centroid size of the pelvis as a proxy for body size. We have verified that centroid size is highly correlated with svl (see lines 154-156) and are therefore confident that this does not introduce any biases.

177: unclear what is meant by “unique morphologies”

Response: We have rephrased this section and avoid mentioning this term in the revised version. Please see our response to the next comment for the exact rephrasing.

181-185: it might be more straightforward to briefly describe anole phylogeny: anoles colonized the Caribbean twice from the mainland, once giving rise to the species in the Greater and northern Lesser Antilles, the second time giving rise to the clade in the southern Lesser Antilles

Response: We agree and have included a sentence that provides an overview of the colonization history of the entire *Anolis* clade.

Original: *Since the adaptive radiation in the Caribbean is considered to have generated unique morphologies, we contrasted three major groups;*

Revised (lines 195-199): *The Anolis clade originated on the mainland, but has colonized Caribbean islands in two waves, one to the Southern Lesser Antilles, and one to the Greater and Northern Lesser Antilles, and the latter gave rise to a re-colonization of the mainland. Since the Southern Lesser Antillean clade contains only few species, we divide Anolis lizards into three major groups;*

248-252: Bonferroni correction is overly conservative, as illustrated by the results reported here. With 105 tests, one would expect 5.25 false positives, yet Bonferroni correction disqualifies 11 and 15 positive results. This is an excessive loss in power. I would either delete the Bonferroni correction or, minimally, point out the tradeoff between Type I and Type II error.

Response: We agree that a Bonferroni correction after permutations is conservative and have therefore followed the advice and do not apply this correction. Our main conclusion remains unchanged by this modification.

Sentence removed (lines 273, 274): *None of these differences between pairs of species are upheld after Bonferroni correction for multiple testing.*

327-329: this statement creates a straw-man of what the authors label the “plasticity first” hypothesis. That hypothesis never claimed that the major differences in limb length among ecomorphs is the result of plasticity, nor, contrary to the authors’ recent work, did it suggest that plasticity played a major role in ecomorph differentiation. The results reported here do not refute the possibility that plasticity can alter limb length to some extent, as the authors themselves showed in their previous work. Moreover, a large body of research, some of it cited in this paper, indicates the adaptive value of minor changes in limb length; given this, it is an over-reach to conclude that plasticity is a “rather unimportant mechanism for the generation of adaptive variation.”

Response: The reviewer raises an important issue here. First of all, we would like to point out that we did not state that *plasticity* is a rather unimportant mechanism for generating adaptive variation, but that *post-hatching regulation of limb growth* is rather unimportant in this context (line 349). This statement follows from our results showing that relative limb length differences as well as limb proportions are established at hatching. We modulated this statement in two respects to address the reviewer’s criticism: first, we qualify our statement about *post-hatching regulation of limb growth* and merely state that it might be less important relative to pre-hatching stages, and second, we include the caveat that we did not study post-hatching stages, and can therefore not draw any inference (other than extrapolating from hatching to adult morphology). In addition, we would like to point out that we discuss the potential for plasticity to influence adaptive variation during pre-hatching stages (lines 357-360).

Original: *This suggests that post-hatching regulation of limb growth is a rather unimportant mechanism for the generation of adaptive variation.*

Revised (lines 351-354): *This suggests that post-hatching regulation of limb growth might be a less important mechanism for the generation of adaptive variation than pre-hatching mechanisms. However, since our study did not include post-hatching stages, we cannot infer if growth trajectories in juveniles differ between *Anolis* species.*

355: the most “extreme” morphologies are not necessarily exhibited by “non-ecomorph” species. Incidentally, some have criticized the term “non-ecomorph” as defining species by what they aren’t; for that reason, “unique” has been used more recently, which also emphasizes the evolutionary convergence aspect of the ecomorphs, a point that could have been made clearer when “ecomorphs” were introduced

Response: This is a very good point. While some non-ecomorph species (e.g., *A. porcus*) are more extreme (i.e., more disparate in morphology) than ecomorph species, this is not consistently the case. We have therefore followed the reviewer’s suggestion and replaced ‘extreme morphologies’ with ‘unique morphologies’ (line 386).

393-394: subject-verb tense disagreement

Response: Thanks you for pointing this out. We have corrected this mistake.

Reviewer #2 (Comments to Author):

Feiner et al. measured proportions in limb bones of 15 species of anoles from hatching to adulthood, and from adult anoles from another 267 species. They compared developmental allometric growth and evolutionary allometric growth in these datasets to see if they are related. If the allometries were ‘aligned’ they propose that would indicate early developmental integration channels diversification in the clade. Unfortunately, they did not measure the stage where adjacent elements were developmentally integrated and influencing proportions, so the question remains unanswered at that level. They wanted to determine when differences in proportions arise, with the stated goals...

“when in ontogeny do species specific relative limb length is established”

“do the different proportions emerge through differential rate of growth of individual bones or different size of limb anlagen?”

However they are not able to answer these questions because they do not measure anlagen at the time of formation; they only measure later development once all the elements are formed. Limb and digit elements form in sequence. The embryos were not evaluated prior to stage 11. By this time, the final phalanx is formed and the interdigital tissue is regressing, and (they showed) the final proportions are already established. So, as they discovered, this paper is actually comparing postmorphogenetic growth allometries with evolutionary patterns of proportion changes.

Response: We thank the reviewer for her/his time in reviewing our manuscript and are grateful for the constructive criticism. It is indeed correct that our results show that (some) species-level differences occur already at a stage prior to the earliest stages in our dataset. However, this is not a weakness of our study, but simply reflects the reality of developmental biology. The elements that we are concerned with (bone elements) cannot be easily tracked at earlier developmental stages than the ones we are including since they simply do not exist *as such* before that. Long bones arise from condensations of mesenchymal cells, which do not have a defined shape and can therefore not be quantified with our methodology before a certain point in development (which we located to stage 11). To alleviate this problem, one could measure the size of the entire limb bud (see Andrews & Skewes, 2017, *Evol Dev*, DOI: 10.1111/ede.12221), but this data could not be used to ask questions about an alignment between ontogenetic and phylogenetic datasets. We believe that applying a different methodology to obtain a more fine-scale resolution of the morphogenetic processes is a promising approach for future studies, but it would not have been suitable for the main purpose of our present study. We have included this aspect in our revised Discussion.

Sentence added (lines 373-375): *Further pinpointing the exact emergence of species-specific differences will require a modified methodology that does not rely on quantifying cartilaginous structures (e.g., [49]).*

The authors did answer the question of whether embryonic postmorphogenetic (aka fetal) allometries match evolutionary allometries - and the answer is that they did not. This makes sense, and it would be useful to expand more on this explanation in this large morphologically diverse sample of anoles. These kinds of datasets are extremely rare and laborious to obtain, so this would be an excellent addition to the literature on allometry and growth. For example, their mention of the lack of support for "plasticity-led" evolution is also useful in this context. There may be other reasons to consider why adult proportions would be stabilized so early in development.

Response: We thank the reviewer for pointing out the significance of our findings and for suggesting to expand the discussion. We refrained from adding more elements to the discussion since we believe that it provides a balanced and thorough treatment of the main discussion points as it is. However, we took this opportunity to clarify the section on plasticity-led evolution to prevent over-interpretation of our current findings and to explain their relationship to our previous work on the same system.

Original: *This conclusion is consistent with previous refutation of the ‘plasticity-first’ hypothesis [12], which postulates that adaptive evolution of the locomotor skeleton in*

Anolis proceeded via modification of the plastic responses to mechanical stress imposed by running versus climbing [41, 46, 47].

Revised (lines 354-359): Nevertheless, this conclusion is consistent with a previous study [12], which showed that modification of the plastic responses to mechanical stress imposed by running versus climbing contributed little to adaptive evolution of the locomotor skeleton in *Anolis*.

Earlier in development, the distal phalanges will be still forming while the proximal elements will be fully formed and growing at a slower rate to 'arrive' at the final proportions just after the tip forms.

If the authors wish to understand how limb and digit proportions arise and are developmentally regulated, they should explore the literature on joint patterning. In the digits, the individual skeletal elements are not formed as individual condensations, but rather the joint positions form by downregulating the cartilage pathway within the condensation, creating a particular joint position. It is at that time when the proportions are established. (e.g. Kavanagh et al., 2013 PNAS, Supplementary 2A). The allometries in this brief stage are where developmental regulation of proportion occurs.

Response: We thank the reviewer for this insight into the developmental biology of phalangeal elements. In particular the different developmental mechanisms of long bone versus phalangeal formation are relevant to our findings, and we have added this insight into the discussion.

Sentence added (lines 391-394): Interestingly, the mechanisms of bone element morphogenesis differ between long bones, metapodials and phalangeal elements [52], which suggests that changes in limb length and proportions can result from a variety of different mechanisms, providing promising targets for future studies.

To make the data more accessible to researchers studying phenotypes, I wish more intuitive figures were prepared for the main manuscript including explanations that refer back to phenotypes and individual proportions to help with understanding. The supplementary helps a lot though. A figure with images of the cleared-and-stained limbs in the developmental series would be a great addition.

Response: We agree with this point. However, due to hard constraints on the page limit set by the journal, we cannot expand the figures in the main part of the manuscript. Instead, we have added a new supplementary figure (S2) that presents illustrations that help the reader to grasp the nature of our study object. The new supplementary figure S2 illustrates the anatomy of fore- and hindlimbs in embryos and adult *Anolis* lizards and the specific measurements which were taken in this study.

Legend added (supplementary figure S2): Figure S2. Limb morphology in embryonic and adult *Anolis* lizards. Limb bone length and body size (i.e., spine length) were measured in cleared and stained embryos (a). Measured elements are indicated in turquoise in the lower panels (a). The same limb bones were measured using micro-CT scans of adult specimens, indicated by red lines (b).

“343 both bivariate and multivariate analysis of limb allometry suggest that the adaptive features
344 that define *Anolis* habitat specialists are to a large extent established at the earliest phase of
345 limb development, prior to the emergence of cartilaginous condensations.”

—Rephrase to “...prior to or during the emergence...”

Response: We have modified this phrasing according to the reviewer’s suggestion (line 373).

Appendix B

Below is the report of our manuscript revision. We show sentences taken from the manuscript in italic, and parts newly inserted into the manuscript as underlined. Page and line numbers refer to positions in the document with tracked changes.

REVIEWER COMMENTS

Associate Editor (Comments to Author):

The reviewers agree that the revised version is very good, and only a few very minor points remain to be addressed. Most importantly, please indicate somewhere (in the supplementary info. section of the manuscript and/or the project description on MorphoSource) that a list of DOI's is available and where it is located. Currently there's no indication that the DOI's are on Dryad. In addition, you may wish to clarify anatomical terms (e.g., "long bones") as suggested by Reviewer 1.

Response: We thank the Associate Editor for the overall positive evaluation of our manuscript revision and for accepting it for publication. Regarding the linking of this manuscript with the CT-scans deposited at MorphoSource, we have added a specific reference to the 'Data accessibility statement' at the mc.manuscriptcentral.com portal. The statement now reads like this:

Raw CT-scans of adult Anolis specimens are deposited at MorphoSource under the project ID P1059, title 'Anolis sp.'. Code and datasets, including a list of DOIs of individual CT-scans deposited at MorphoSource, are available at Dryad (<https://doi.org/10.5061/dryad.jsxksn08d>).

Regarding the remaining criticism raised by Reviewer #1, we refer to our response below.

We believe that our revised manuscript fully satisfies the outstanding criticisms and is now ready for publication.

Reviewer #1 (Comments to Author):

The authors have done a nice job of revising their manuscript, which was already quite good. I only have two comments:

376-382: it may be relevant to note that *angusticeps* and *sagrei* are distantly related, whereas by design, Sanger et al. compared pairs of closely related species that differed in limb length.

Response: We do not fully agree with this statement. While one species pair investigated by Sanger *et al.* was indeed a relatively closely related species pair (*A. lineatopus* versus *A. graham*; 21 million years divergence time), the other species pair (*A. sagrei* versus *A. carolinensis*; 44 million years divergence time) was as distantly related as the species pair studied by Wakasa *et al.* (*A. sagrei* versus *A. angusticeps*; also 44 million years divergence time; divergence time estimates according to Poe *et al.*, 1017, *Syst Biol*). Therefore, we think it is not justified to discuss further the details of these two studies with respect to their study design and species selection.

402, paragraph: I continue to believe that the authors are failing to own up to the results of their study. The finding that evolutionary allometry fails to mirror ontogenetic allometry—indeed, is in the opposite direction—suggests that, in fact, development is not constraining evolution, that evolution is not occurring along lines of least resistance. The authors may have set out to find support for this hypothesis, but I feel they are not being upfront in acknowledging that the results would seem to falsify that hypothesis. If nothing else, the authors should more explicitly discuss how these results sit in the context of their initial hypotheses.

Response: We understand the reviewer’s position, but we do not actually agree that it follows from these results that *‘development is not constraining evolution’* (to quote the reviewer). What we do show is that *‘evolutionary diversification of limb morphology has not followed the ontogenetic growth trajectory of limb bones’* (section Conclusion, lines 435, 436). That this implies that ontogenetic growth processes do not act as developmental constraints is discussed at some length in our manuscript – in this sense we *‘falsify that hypothesis’* and hence agree with the reviewer. However, we disagree with the reviewer in interpreting these results as showing that *‘development is not constraining evolution’*. In our opinion, thinking of developmental constraints as a force that can either be present (preventing the evolution of fit phenotypes) or absent (natural selection has free range and everything is possible) can be counterproductive since there can be no phenotype without development. Thus, we advocate replacing this dichotomous thinking with a concept of developmental bias that reflects that developmental processes will always make some solutions more likely than others. We recognize that our take on this issue has been implicit in the discussion of the manuscript, and we chose to emphasize this point in the revised manuscript (see also our response to the next comment):

Sentence added (lines 394-396): *These mechanisms may themselves exhibit developmental bias, which has been suggested to explain why the distal-most phalanges show the greatest variability and evolvability in birds [59].*

We recognize that this is a conceptual debate that cannot be settled in the context of our manuscript, but we prefer to present our interpretation of the results rather than making stronger claims about the role of developmental bias in adaptive evolution of limbs.

Along those lines, the back half of the paragraph, starting on line 410 “this is consistent with...” does not make sense to me. I do not see how that material has any bearing on the preceding material, much less being consistent with it. On re-reading a third time, I guess the point has to do with “distal” vs. “long” bones. The authors should make this more explicit for those who are not anatomists. Regardless, since the contrasting pattern of positive versus negative allometry pertains to the long bones, the material on distal points seems tangential.

Response: We agree with the reviewer that this section might have been difficult to follow and we therefore restructured this paragraph.

Originally: *In particular, the proportional increase in the length of long bones was positive in embryonic development, but tended to be negative across species. This is consistent with the observation that differences in the length of distal bones account for much of the variation in relative limb length of different Anolis ecomorphs [25].*

Revised (lines 389-399): *In particular, the proportional increase in the length of long bones (proximal elements) was positive in embryonic development, but tended to be negative across species. Similarly, Anolis phalangeal (distal) elements exhibit patterns of co-evolution with other limb bones that are distinct from their co-development. As explained above, these results suggest that the evolution of the Anolis limb involved modifications of developmental mechanisms early in development. These mechanisms may themselves exhibit developmental bias, which has been suggested to explain why the distal-most phalanges show the greatest variability and evolvability in birds [59]. Distal elements do appear to be particularly evolvable in Anolis as well. Across the phylogeny, differences in the length of distal bones account for much of the variation in relative limb length of different Anolis ecomorphs [25].*

Sentence deleted (lines 403, 404): *The distal-most phalanges show the greatest variation in birds [59].*

Sentence deleted (lines 407-409): *In contrast to the morphology of the cichlid jaw, however, Anolis phalangeal elements exhibit patterns of co-evolution with other limb bones that are distinct from their co-development.*

Reviewer #2 (Comments to Author):

The revisions are sufficient.

Response: We thank the reviewer for her/his time in reviewing our manuscript and are grateful for the positive evaluation.